# Evaluating Healthcare-Associated Infections in Public Hospitals: A Cross-Sectional Study

**DOI:** 10.3390/antibiotics12121693

**Published:** 2023-12-02

**Authors:** Daniela Iancu, Iuliu Moldovan, Brîndușa Țilea, Septimiu Voidăzan

**Affiliations:** 1Doctoral School, University of Medicine, Pharmacy, Sciences and Technology George Emil Palade of Tîrgu Mureș, 540141 Târgu Mureș, Romania; daniela.iancu@umfst.ro; 2Discipline of Public Health and Health Management, University of Medicine, Pharmacy, Science and Technology George Emil Palade of Targu-Mures, 540141 Târgu Mureș, Romania; 3Department of Infectious Disease, University of Medicine, Pharmacy, Sciences and Technology George Emil Palade of Tîrgu Mureș, 540141 Târgu Mureș, Romania; brindusa.tilea@umfst.ro; 4Department of Epidemiology, University of Medicine, Pharmacy, Sciences and Technology George Emil Palade of Tîrgu Mureș, 540141 Târgu Mureș, Romania; septimiu.voidazan@umfst.ro

**Keywords:** hospital-acquired infections, colonization, infection, risk factors, antibiotics

## Abstract

Background: Hospital-acquired infections (HAIs) pose a significant danger to global public health, mainly because their numbers are growing exponentially each year. Additionally, the rise of bacterial strains resistant to current treatment options further exacerbates this threat. This study aimed to examine the occurrences of HAIs identified in public hospitals at the county level. Methods: We conducted a cross-sectional study utilizing data provided to the Mures Public Health Directorate from all the public hospitals within the studied county. We examined HAIs reported during the period spanning from 2017 to 2021, which amounted to a total of 4603 cases. Results: The medical departments reported the highest prevalence of HAIs at 48.25%. The most common infections included enterocolitis with *Clostridioides difficile* (32.61%), COVID-19 (19.83%), bronchopneumonia (16.90%), sepsis, surgical wound infections, and urinary tract infections. The five most frequently identified pathogens were *Clostridioides difficile* (32.61%), SARS-CoV-2 (19.83%), *Acinetobacter baumannii* (11.82%), *Klebsiella pneumoniae* (9.58%), and *Pseudomonas aeruginosa* (7.95%). *Acinetobacter baumannii* was the predominant agent causing bronchopneumonia, while *Klebsiella pneumoniae* was the leading cause of sepsis cases. *Escherichia coli* was the primary agent behind the urinary tract infections, and *Staphylococcus aureus MRSA* was identified as the main etiology for wound infections and central catheter infections. Throughout the study period, there was a significant rise in *Clostridioides difficile and Gram-negative* bacteria prevalence rates. Conclusions: This study identified increased *Clostridioides difficile* in HAI cases during COVID-19, highlighting the need for careful antibiotic use and emphasizing the growing challenge of multi-resistant strains in post-pandemic state hospitals.

## 1. Introduction

HAIs, also known as “*nosocomial infections*” occur in patients receiving medical care in hospitals or other healthcare facilities, including chronic care units and retirement homes. In recent years, these infections have posed a significant threat to public health, leading to increased mortality and morbidity. Additionally, there are challenges in limiting and treating these infections effectively [1].

In Europe, the most recent data from the World Health Organization indicate an annual occurrence of 8.9 million HAIs in acute and long-term care facilities. The majority of these cases are reported in intensive care units (ICU), affecting up to 30% of patients in high-income countries and two to three times more in low-income nations. The statistics also reveal that 23.6% of all hospital-treated sepsis cases are healthcare-associated, with a mortality rate of 24.4% among patients affected, rising to 52.3% for those treated in intensive care units. Moreover, there has been an emergence of multidrug-resistant bacteria, including *Vancomycin-resistant Enterococci* (VRE) and *Staphylococcus aureus MRSA*, as Gram-positive agents, as well as *Pseudomonas aeruginosa, Klebsiella pneumoniae*, and *Enterobacter* spp. as Gram-negative pathogens [2]. 

In Romania, HAIs remain underreported, with an official prevalence of 0.2–0.3%, contrasting with the 7.1% prevalence across Europe. A 2018 study on HAI prevalence in Europe, based on two prevalence surveys, indicated a rate of 2.6% in Romania (95% Confidence Interval: 1.7–4.0) [3,4]. The primary issue contributing to these discrepancies in prevalence comparisons is the lack of proper reporting of these infections. In 2016, Romania established Order No. 1101/2016 [5], endorsing standards for the surveillance, prevention, and containment of HAIs within healthcare facilities. The adopted measures included rigorous adherence to hand hygiene protocols involving water, soap, and alcohol-based antiseptics, the utilization of protective attire such as gloves, surgical masks, and disposable gowns, the prudent management of medical equipment, and the meticulous administration of parenterally administered drugs. Following the implementation of this directive, a noticeable surge in the reporting rate of HAIs was observed; however, their prevalence remains significantly below the European average. Common pathologies associated with HAI include mechanical ventilation-associated pneumonia and urinary tract infections in ICUs, wound infections in surgical units, and intestinal tract infections involving *Clostridioides difficile*, which can compound other HAIs due to prolonged antibiotic treatment [6,7]. 

The aim of this study was to analyze the cases of HAIs identified in public hospitals in Mures county, Romania, through case reports sheets, as these sheets are used to ensure that HAIs are reported according to Romanian legislation.

## 2. Results

The patients diagnosed with HAIs had an average age of 60.27 ± 21.67 years (mean ± standard deviation), ranging from a minimum of 3 weeks to a maximum of 99 years. Out of the total 4603 cases, 2454 (53.31%) were males, and 46.69% were females. In terms of residence, 54.16% were from urban areas, and the remaining 45.84% were from rural regions.

HAIs were most prevalent in medical sections (48.25%), followed by ICU wards (32.28%), surgical units (17.18%), neonatology wards (1.39%), and the EDs (0.90%). Regarding the types of HAIs reported, enterocolitis with *Clostridioides difficile* was the most common at 32.61%, followed by COVID-19 (19.83%), bronchopneumonia (16.90%), sepsis (9.04%), urinary tract infections (9.04%), surgical wound/tissue infections (8.30%), central catheter infections (2.04%), and other types of infections (2.24%). Other infections included meningitis, external ventricular drainage infections, otitis, phlebitis, and influenza-type A/B infections. Among the cases reported, 92.48% originated from the reporting hospital, and HAI was considered the possible cause of death in only 7.06% of 1176 fatal cases (Table 1).

The majority of the reported HAIs observed in medical sections (2221 cases) were primarily due to *Clostridioides difficile* (42.05%), followed by SARS-CoV-2 (26.97%). In ICU units, *Clostridioides difficile* infections were predominant (27.79%), followed by *Acinetobacter baumannii* (15.34%), *Pseudomonas aeruginosa* (13.53%), and *Klebsiella pneumoniae* (9.96%), with *Escherichia coli* being the least frequent at 3.57%. In the surgical wards *Acinetobacter baumannii* was the most common pathogen (22.76%), followed by *Clostridioides difficile* (18.33%) and *Staphylococcus aureus MRSA* (13.65%) among all nosocomial surgical cases. Neonatology departments identified *Escherichia coli* as the primary etiological agent (25%), closely followed by *Staphylococcus aureus MRSA* (20.31%). The EDs reported the fewest HAI cases (0.9%), though those that were reported were most often attributed to SARS-CoV-2 (26.83%) and *Staphylococcus aureus MRSA* (21.95%) (Figure 1).

In terms of infection types, enterocolitis was the most common, being exclusively caused by *Clostridioides difficile* in 100% of cases. Among the cases of bronchopneumonia, *Acinetobacter baumannii* was the predominant etiological agent, accounting for 36.15%, followed by *Klebsiella pneumoniae* (18.24%) and *Pseudomonas aeruginosa* (18.21%). Besides other germs, *Escherichia coli* was the primary cause of urinary tract infections in 23.38% of cases, with this infection type being most frequently reported in medical wards and ICU units. *Staphylococcus aureus MRSA* was predominantly isolated in central catheter infections (29.72%) and surgical wound infections (27.49%). The main sources of infection for sepsis were *Klebsiella pneumoniae* (28.33%), *Acinetobacter baumannii* (20.88%), and *Pseudomonas aeruginosa* (19.01%) (Figure 2).

In 1176 cases, death was reported as an outcome, with *Clostridioides difficile* and *Acinetobacter baumannii* identified as the etiological agents in nearly half of the deaths. A total of 36.98% of the patients were discharged with improved status. Among these cases, *Clostridioides difficile* was also the most frequently isolated agent (571). A total of 481 patients were successfully treated and discharged as cured, with *Clostridioides difficile* being responsible for almost half of these cases (41.08%). Data on the discharge status of patients were unavailable in 1002 cases (Figure 3).

The overall prevalence of HAIs was 0.87%, accounting for 4603 out of 530,181 cases of patients discharged from the studied hospitals during the study period. In the pre-pandemic years of 2017–2019, the three most prevalent causative agents were *Clostridioides difficile* at 0.13%, *Acinetobacter baumannii* at 0.09%, and *Klebsiella pneumoniae* at 0.08%. Notably, with the onset of nosocomial infections related to SARS-CoV-2, it claimed the second position. Consequently, the prevalence of HAI with SARS-CoV-2 during the 2020–2021 period was calculated to be 0.48%. Considering the total prevalence over the 5-year span, *Clostridioides difficile* emerged as the most prevalent etiological agent at 0.28%, followed by SARS-CoV-2 at 0.17%, while *Escherichia coli* was the least frequent agent, causing HAIs with a prevalence of 0.03%. (Table 2). 

In 2017, the prevalence of HAIs caused by *Clostridioides difficile* was calculated to be 0.10%. During the study period, a significant increase was observed, reaching 0.80% in 2021. Similarly, there has been a notable rise in HAIs attributed to *Gram-negative* bacteria. The prevalence of *Acinetobacter baumannii* infections, for instance, escalated from 0.05 in 2017 to 0.14 in 2021. Additionally, *Klebsiella pneumoniae* and *Pseudomonas aeruginosa* exhibited an increasing trend in HAI prevalence, reaching 0.12% and 0.11%, respectively, by 2021. The prevalence of HAIs caused by *Staphylococcus aureus MRSA* and *Escherichia coli* displayed a relatively consistent trajectory in the initial four years of the study, only to experience a significant upward shift in 2021, with Staphylococcus at a prevalence of 0.08% and Escherichia coli at 0.06% (Figure 4).

## 3. Discussion

This study monitored HAI cases over a span of 5 years, during which 4603 cases were documented. The average age of the participants ranged from 60 to 65 years, with ages spanning from 3 weeks to 99 years. Older individuals constituted the majority of patients requiring medical services. They form a vulnerable demographic due to their diminished immune system, influenced by both age-related factors and underlying chronic illnesses [8]. Typically, patients diagnosed with HAIs exhibit multiple concurrent health conditions, making them susceptible to a general decline in immune competence. Prolonged or repeated hospital stays increase the likelihood of HAI, occurring not only in ICUs but also in various medical and surgical units [9]. 

All instances were documented from seven public hospitals, establishments inherently more susceptible to the proliferation of pathogens causing HAIs when compared to private institutions. Firstly, state-run hospitals accommodate a significantly larger number of patients than private medical facilities. They have a higher quantity of medical departments and limited space, making it challenging to create isolation rooms [10]. Secondly, financially, public hospitals lack the resources of their private counterparts. Consequently, ensuring an equivalent level of cutting-edge equipment or a continuous supply of cleaning agents and disinfectants remains unfeasible [11]. Conversely, private hospitals might report fewer HAI cases due to potential underreporting. This underreporting could stem from their desire to avoid negative publicity, subsequently reducing patient attendance [12]. 

In this study, it was noted that the majority of HAI cases occurred in medical departments. Before 2020, the studied hospitals primarily reported nosocomial infections in the ICU. However, during the onset of the COVID-19 pandemic, a significant shift was observed, with most HAIs being recorded in medical wards, followed closely by ICUs. In 2020, due to the surge in *SARS-CoV-2* cases and the rise in critical cases, Romanian hospitals found themselves with limited space in their ICUs. Consequently, medical departments, particularly those specializing in infectious diseases and pneumology, had to accommodate a substantial number of patients. The large number of patients admitted in this period of time resulted in a decrease in isolation options for patients vulnerable to infection. Also, despite the pandemic’s emphasis on preventive measures utilizing protective sanitary materials, there were intervals when our nation encountered shortages of these materials, potentially leading to inappropriate usage by auxiliary healthcare personnel attending to patients. Consequently, the emergence of *SARS-CoV-2* has amplified the risk of nosocomial infections.

Additionally, there was a shift in the predominant type of HAI. While bronchopneumonia caused by *Acinetobacter baumannii* was the most common type between 2017 and 2019, starting from 2020, enterocolitis associated with *Clostridioides difficile* became the most frequently reported infection. Consequently, over the five-year study period, enterocolitis stood out as the most prevalent infection, being documented in a total of 1501 cases, with 62.23% of these cases emanating from medical wards. 

*Clostridioides difficile* emerged as the most frequently detected pathogen responsible for HAIs, constituting 1501 out of 4603 cases. Notably, compared to the pre-pandemic years (when *Acinetobacter baumannii* was predominant), *Clostridioides difficile* exhibited a significant surge in its prevalence rate, from 0.10% in 2017 to 0.80% in 2021. Particularly striking was the rise coinciding with the emergence of the SARS-CoV-2 virus, which accounted for nearly half of all HAI cases in 2021. Indeed, the prevalence of *Clostridioides difficile* infections in the 2020–2021 period surpassed those reported between 2017 and 2019. Due to the onset of COVID-19, stringent measures were enforced to enhance hand hygiene, mandate protective equipment, and encourage social distancing. The objective behind these measures was to diminish the transmission of infectious diseases, as substantiated in certain facilities [13]. Nevertheless, it is essential to acknowledge that throughout the pandemic, in-hospital patients displayed several risk factors for *Clostridioides difficile* infection, such as prolonged hospital stays, antibiotic and steroid treatments, advanced age, and compromised immune status. Some studies have revealed a reduction in *Clostridioides difficile* infections following the introduction of COVID-19 control measures when compared to the preceding years, even in the face of heightened antibiotic usage [14].

On the other hand, various studies have indicated a rise in *Clostridioides difficile* infection rates [15]. A recent extensive study across multiple centers disclosed an 11% surge in HAIs associated with *Clostridioides difficile* during the pandemic [16]. An analysis covering the COVID-19 era up to the beginning of 2021 identified elevated *Clostridioides difficile* rates during the peak periods of the COVID-19 pandemic [17]. In line with our findings, a recently published paper indicated a trend similar to that of *Clostridium difficile* infections over a 5-year period (2018–2021). It revealed that the infection incidence rate increased from 5.67 +/− 0.35 before the onset of the pandemic to 8.06 +/− 0.41 in the 2020–2021 period (95% confidence interval) [18].

Several factors could explain this trend. Firstly, during the pandemic, healthcare facilities faced overcrowding, making it challenging to isolate potential community cases effectively. Consequently, bacterial transmission among patients became inevitable. Furthermore, managing antibiotic treatment for COVID-19 patients presents challenges due to similarities between bacterial pneumonia and moderate to severe COVID-19 cases. Although current guidelines recommend using empiric antibiotics only when bacterial infections are suspected in moderate COVID-19 cases, they are routinely recommended for severe COVID-19 cases. Distinguishing between advancing COVID-19 illness and bacterial co-infection or superinfection is challenging. The decision for empiric antibiotic treatment might be influenced by experiences from *Influenza* cases, where bacterial co-infection rates range from 11% to 35% [19]. During the pandemic, our hospitals commonly utilized prophylactic antibiotic therapy for these patients. The use of broad-spectrum antibiotics is associated with the induction of *Clostridioides difficile* infection, and studies indicate that antibiotic treatment for bacterial infections in hospitalized COVID-19 patients leads to the depletion of beneficial commensals and the proliferation of opportunistic pathogens, including *Clostridioides difficile* [20]. 

Apart from enterocolitis, the leading types of nosocomial infections included COVID-19, bronchopneumonia, urinary tract infections, sepsis, and surgical wound infections. Bronchopneumonia was notably frequent in ICU cases, with *Acinetobacter baumannii*, *Pseudomonas aeruginosa*, *Klebsiella pneumoniae*, and *Staphylococcus aureus MRSA* being the primary causative agents and especially prevalent in ICU sections. The extensive use of tracheal intubation and mechanical ventilation for critically ill patients further heightens the risk for this condition already susceptible among patients [21]. In line with our findings, a cohort study conducted by Trivedi et al. [22] revealed that the most common isolates in ICU nosocomial pneumonia were *Pseudomonas aeruginosa* (55%), *Acinetobacter baumannii* (20%), *Staphylococcus aureus* (14.5%), and *Klebsiella pneumoniae* (7.5%). A limited number of nosocomial lower respiratory tract infections were linked to *Escherichia coli.* Aspiration and mechanical ventilation were identified as risk factors for this condition [23]. 

Our study highlighted a gradual increase in the prevalence of HAIs caused by *Acinetobacter baumannii*, *Pseudomonas aeruginosa*, and *Klebsiella pneumoniae* strains, particularly in the last two years of the study period, 2020 and 2021. Numerous recent scientific articles have demonstrated a significant rise in HAIs associated with these strains since the onset of the COVID-19 pandemic [24,25]. A review study published in June 2023 showed an overall increase in drug-resistant bacterial infections involving *Klebsiella pneumoniae*, *Pseudomonas aeruginosa*, *Acinetobacter baumannii*, and other germs, correlating with the heightened use of antibiotics in COVID-19 admissions [26]. An interesting observation stems from a recent study that analyzed the incidence of HAIs in patients with and without COVID-19. The findings of this specific study suggest that either the disease or its treatment appears to selectively heighten the susceptibility of the COVID-19 population to HAIs [27]. 

Following this article, additional comprehensive studies are essential to assess the current antibiotic resistance of the germs reportedly causing HAIs in our region and to analyze if the COVID-19 pandemic changed the antibiotic resistance of these strains. 

### Limitations of the Study

The current study has some limitations, including its retrospective nature and the fact that data analysis was conducted solely at the county level, making it insufficient to draw valid comparisons with other regions in Romania. Moreover, since the study analyzed only the reported cases of HAIs, the data utilized were incomplete with respect to the risk factors and comorbidities of the patients studied. Further studies are needed to elucidate how the factors described may have contributed to the results reported in this paper. Additionally, the discharge statuses for a significant number of cases remain unknown.

## 4. Materials and Methods

We conducted a cross-sectional study analyzing cases reported by public hospitals in Mures County (Romania) to the Mures Public Health Directorate. The inclusion criteria were patients admitted between 2017 and 2021 who were diagnosed with an HAI according to the definition mentioned in Order No. 1101/2016. The establishments considered in the study included county clinical hospitals, a cardiovascular disease institute, municipal hospitals, and a city hospital. No cases were reported from private hospitals. The study focused on HAIs reported between 2017 and 2021, totaling 4603 cases. Our data analysis considered patient demographics, hospital departments, diagnoses, surgical procedures, HAI diagnoses, isolation protocols, the pathogens involved in HAI cases, antibiotic resistance, treatment details, patient status upon discharge, and potential links between HAIs and patient deaths. The study encompassed various medical and surgical departments, which were later categorized into ICUs, surgical units (general surgery, plastic surgery, obstetrics and gynecology, orthopedics, neurosurgery, urology), medical units (infectious diseases, cardiology, diabetes, hematology, gastroenterology, palliative care, internal medicine, neurology, nephrology, oncology, pediatrics), neonatology units, and emergency departments (EDs) for statistical analysis. We compiled data on the total discharges from the investigated hospitals, considering each mentioned year. This information was used to compute the prevalence rates of the HAIs reported within this specified time period. Regarding the processing of the data, we received the consent of the Mures Public Health Directorate, and confidentiality and anonymity were maintained. 

### 4.1. HAI Identification and Diagnosis in Romania

On 30 September 2016, the Romanian Minister of Health issued Order No. 1101/2016 [5], approving regulations for the supervision, prevention, and containment of infections associated with medical care in healthcare facilities. According to this document, an HAI is defined as an infection acquired in a healthcare setting (either public and private), and HAIs encompass all infectious diseases that are identifiable clinically and/or microbiologically. There must be epidemiological evidence indicating contraction during hospitalization or medical/surgical procedures. The incubation period of these infections is linked to the duration of medical care received by the patient in these facilities, regardless of whether symptoms appear during the hospitalization period. To diagnose HAIs, it is crucial to establish that the infection resulted from hospitalization or medical care in healthcare settings and that the infection was not present during the incubation/initial phase of the disease at the time of hospitalization or medical procedure.

When a physician suspects an HAI, they promptly notify the hospital’s HAI prevention department, following the European Union’s specified case definitions. The microbiology lab investigates suspected cases within 24 h. Upon germ identification, bacteriology notifies the Public Health Directorate and relevant medical departments, especially when multidrug-resistant germs are isolated. The attending physician collaborates with an epidemiologist to assess the case clinically and classify it as confirmed, denied, or colonization based on definitions. The attending physician completes paperwork and files the HAI case statement, and the HAI diagnosis is logged in the patient’s clinical records. The HAI diagnosis is recorded via a general clinical observation sheet and computer system at the patient’s discharge or transfer. The responsible physician ensures data accuracy. Suspected cases trigger daily electronic reports, while confirmed cases are reported weekly to the HAI prevention department. At the hospital level, the department maintains centralized records and prepares epidemiological investigations, implementing measures to prevent pathogen transmission. Monthly, quarterly, and annual reports on HAI incidence, categorized by sections and infection types, are submitted to the hospital’s Steering Committee and the county’s Public Health Directorate.

### 4.2. Statistical Analysis 

We utilized Statistical Package for Social Sciences (SPSS) for our statistical analysis. Normally distributed quantitative data, such as age, were expressed as mean and standard deviation. The prevalence of HAIs was assessed by determining the percentage of patients with infections among the total number of hospitalized patients during each specified period. Qualitative data were represented as counts and percentages.

## 5. Conclusions

This study conducted a statistical analysis of HAI cases in Mures County, Romania. It identified *Clostridioides difficile* as the most prevalent pathogen reported as a nosocomial infection over a 5-year period, encompassing both pre-pandemic and pandemic times. Notably, a correlation was observed between *SARS-CoV-2* infection and the rise in enterocolitis cases caused by *Clostridioides difficile*. Also, we noted a significant increasing trend in the prevalence of HAIs caused by Gram-negative bacteria. This study underscores the importance of judicious antibiotic use in routine medical practice. Prolonged and excessive usage, or prophylactic administration without substantiated justifications for infectious risk, can contribute to fostering an environment conducive to the heightened prevalence of HAIs.

## Figures and Tables

**Figure 1 antibiotics-12-01693-f001:**
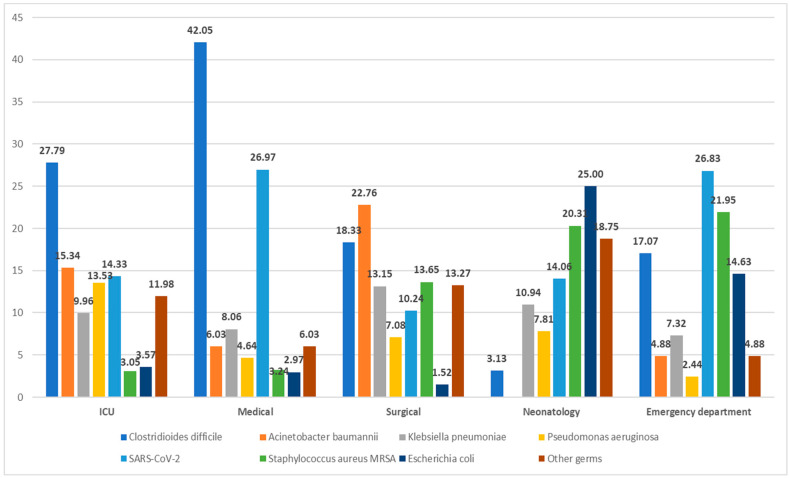
The identification of pathogens according to the specific departments examined.

**Figure 2 antibiotics-12-01693-f002:**
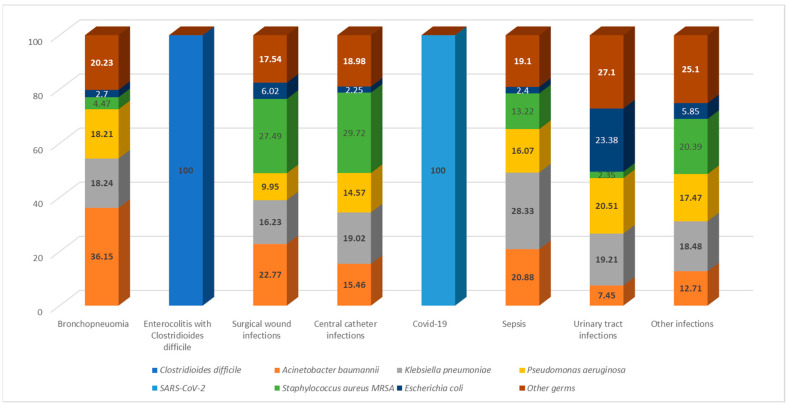
The identification of etiological agents according to the type of HAI.

**Figure 3 antibiotics-12-01693-f003:**
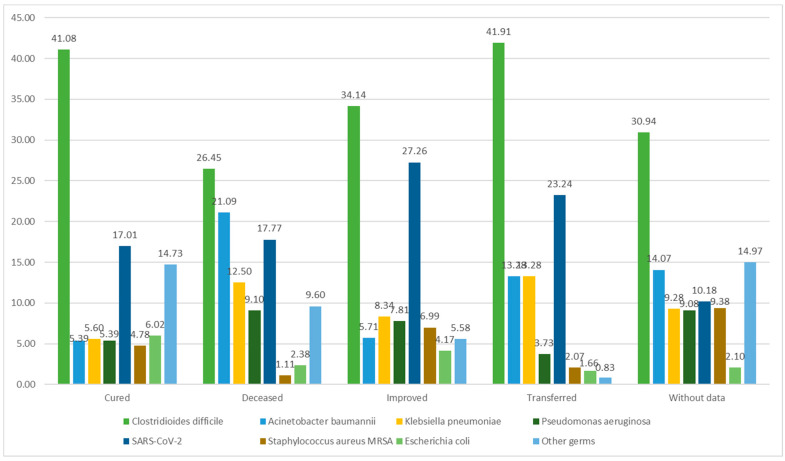
The distribution of pathogens according to the patient’s status upon discharge.

**Figure 4 antibiotics-12-01693-f004:**
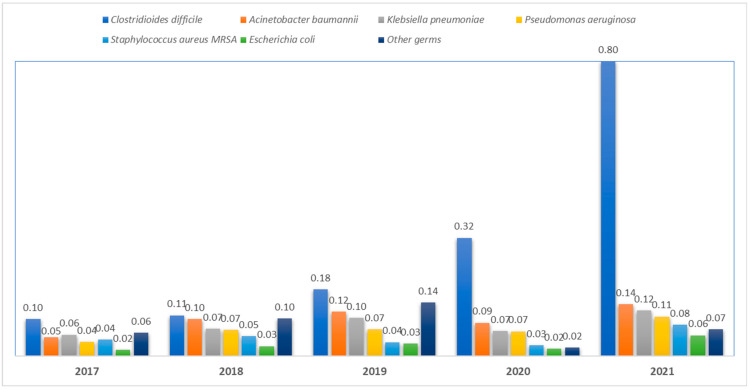
The prevalence of the causative agents throughout the studied timeframe.

**Table 1 antibiotics-12-01693-t001:** Characteristics of the study group.

Caption	Number (4603)	Percentage (%)
Age (mean ± SD)	60.27 ± 21.67	0–99 years
Gender (males)	2454	53.31
Residence (urban)	2493	54.16
Isolated (yes)	2301	49.99
Contacts (yes)	2536	55.09
**Section type**		
Intensive care unit	1486	32.28
Surgical	791	17.18
Medical	2221	48.25
Neonatology	64	1.39
Emergency department	41	0.90
**Types of infections**		
Bronchopneumonia	778	16.90
Enterocolitis with Cl. diff.	1501	32.61
Surgical wound infections/tissue infection	382	8.30
Systemic infections (sepsis)	416	9.04
Central catheter infections	94	2.04
Urinary tract infections	416	9.04
COVID-19	913	19.83
Others	103	2.24
**Origin of the case**		
From the reported hospital	4257	92.48
From another hospital	234	5.08
Chronic/elderly care units	27	0.59
Other types of medical care	85	1.85
Status on discharge		
Cured	482	10.47
Improved	1702	36.98
Deceased	1176	25.55
Transferred to another unit	241	5.24
No data	1002	21.77
**Cause of death**		
Possibly caused by the HAI	325	7.06
Not related to HAI	935	20.31
Unknown	153	3.32
No data	3190	69.30

**Table 2 antibiotics-12-01693-t002:** Prevalence of HAIs pre- and post-pandemic.

**Prevalence of HAIs in 2017–2019**	
**Agent**	**Prevalence (%)**
** *Clostridioides difficile* **	0.13
** *Acinetobacter baumannii* **	0.09
** *Klebsiella pneumoniae* **	0.08
** *Pseudomonas aeruginosa* **	0.06
** *Staphylococcus aureus MRSA* **	0.04
** *Escherichia coli* **	0.03
** *Other germs* **	0.10
**Prevalence of HAIs in 2020–2021**	
**Agent**	**Prevalence (%)**
** *Clostridioides difficile* **	0.55
** *SARS-CoV-2* **	0.48
** *Acinetobacter baumannii* **	0.12
** *Klebsiella pneumoniae* **	0.09
** *Pseudomonas aeruginosa* **	0.08
** *Staphylococcus aureus MRSA* **	0.05
** *Escherichia coli* **	0.04
** *Other germs* **	0.04
**Total Prevalence of HAIs in 2017–2021**	
**Agent**	**Prevalence (%)**
** *Clostridioides difficile* **	0.28
** *SARS-CoV-2* **	0.17
** *Acinetobacter baumannii* **	0.10
** *Klebsiella pneumoniae* **	0.08
** *Pseudomonas aeruginosa* **	0.07
** *Staphylococcus aureus MRSA* **	0.05
** *Escherichia coli* **	0.03
** *Other germs* **	0.08

## Data Availability

Data are contained within the article.

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
