# Peer review of "Evaluating Healthcare-Associated Infections in Public Hospitals: A Cross-Sectional Study"

_antibiotics, 2023, doi:10.3390/antibiotics12121693_

Round 1
Reviewer 1 Report
Comments and Suggestions for Authors
See file attach.

Comments on the Quality of English LanguageAuthor Response
antibiotics-2740346-peer-review-v1
We appreciate your active engagement and considerate recommendations. In alignment with these, we have incorporated the ensuing modifications, which are highlighted in the manuscript:
Introduction
What is the hand hygiene policy to prevent HAI in the hospital?
Within the manuscript, we have incorporated specific information regarding the preventive measures for HAIs as outlined in Order No. 1101, which addresses surveillance, prevention, and containment of HAIs. This includes details on hand hygiene protocols.
Discussions
What is the possible explanation for the high prevalence of clostridium? Is this only because of overcrowding? (p9. line 180) There is a minimum space between hospital beds. What does the author mean about overcrowding? The author should add information and discuss it subsequently. What is the patient's history of antibiotic use?
In accordance with your question, we incorporated into the manuscript potential explanations for the heightened prevalence of HAI involving Clostridioides difficile. In accordance to other reviews that we received, we shortened and revised the entire discussion part. We included more comparisons with those of the authors. We also incorporated some prevalence rates in our study and made some changes in the discussion's over all.
Results
How did the author determine 'Possibly caused by the HAI' (Table 1) in this study?
We made a modification to the manuscript after your evaluation. After we consulted with the attending clinicians, we determined that “possibly caused by HAI” were cases in which the clinical condition of the patients deteriorated considerably after the association of nosocomial infections. The attending physicians determined that HAI was the cause of death. In accordance, we established that in 325 cases, HAI was the cause of death.
Methods
The author should add information about HAI determination to hinder misclassification bias. Is 'The HAI diagnosis (coded Y95)' only the inclusion criteria?
We thank you for your useful suggestions regarding the material and methods section. We made the changes accordingly. We specified the inclusion criteria more clearly as you suggested. After some consideration, the codification method - Y95- was erased from the text.
What the author means about underreporting in p12. Line 325?
As we mentioned above, we made some changes in the discussion part. We thank you for your input, as it made it clear that we need to revise that part.
- 13 Line 388: Qualitative data were represented as counts and percentages. Is percentage a qualitative data?
We made changes in the statistical analysis section, after your prompt observation. We modified “qualitative” to quantitative and we thank you for highlighting the error.
Conclusion
The conclusion is too broad, not from the study's result.
What is the implication of this study for daily practices?
We kindly thank you for your suggestion. We made modifications in the conclusion section, underlining the implication of this study on daily practice. These implications regard the importance of proper antibiotic use, as prolonged and excessive usage, or the prophylactic administration without substantiated justifications for infectious risk, can contribute to fostering an environment conducive to the heightened prevalence of HAI.
Reviewer 2 Report
Comments and Suggestions for Authors
The study focuses on hospital-acquired infections (HAIs) in public hospitals in Mures County, Romania. It is a cross-sectional study using data from 2017 to 2021, covering 4603 HAI cases. Key findings include:
· The most prevalent HAIs were enterocolitis with Clostridioides difficile, COVID-19, and bronchopneumonia.
· The study emphasizes the increasing trend of Clostridioides difficile cases, especially during the COVID-19 pandemic, and highlights the challenges posed by multi-resistant bacterial strains.
· It discusses the importance of appropriate antibiotic use and the need for continued vigilance in managing HAIs, particularly in the context of the pandemic.
· Of the 30% of patients evaluated, only 7% of deaths could be attributed to the HAI
The study's strengths lie in its comprehensive data collection over five years and its focus on a range of HAIs. However, the authors acknowledge limitations such as regional focus and incomplete data on patient comorbidities and risk factors.
I have several comments, which mainly relate to the cross-sectional method and the manuscript's scarcity of details on HAI definitions and data collection. Clear definitions and transparent reporting of data collection methodologies are crucial for reproducibility and reliability. I would encourage the authors to elaborate on the methods of data collection and analysis to bolster the study's credibility.
Material and methods
The authors report a high proportion of HAIs caused by Clostridioides difficile, Covid-19 and Acinetobacter infections which is interesting, but to report the relative percentage of all HAIs as the only metric gives in my opinion relatively little information.
1) I suggest you include prevalence rates, which also aligns better with your use of cross-sectional and prevalence as your method, i.e., number of infections per year and population size. This would greatly improve the quality of this retrospective study. Standardizing the number of HAIs per total number of occupied hospital beds or by 1000 patient days, e.g. quarterly or per annum, is standard epidemiological practice. Such metrics would allow for more nuanced analyses, to benchmark between different hospitals, and better be able to study temporal trends.
2) Regarding Covid-19 epidemiology, one may retrieve much information of interest from open sources. According to downloadable European Covid-19 data from the ECDC database, Romania in 2020-21 registered 1806554 new cases. The region of MureÊ‚ had a 2023 population of 518193 (Wikipedia), out of 19 million in Romania. This indicates a national prevalence per 1000 inhabitants of Covid-19 cases of 0,96 for the pandemic period (1806554 cases / 19000000 inhab * 1000) compared to the prevalence of nosocomial Covid-19 HAI cases which is on average 0,17 in Mures hospitals (913 HAI cases / 518193 inhab * 1000). This only 5-6-fold lower prevalence in hospitals than in the community seems high – but only a calculation of the prevalence rate using the number of bed-days or discharges from the hospitals as denominators, will allow comparison with international prevalence data.
I propose that this would improve the article and allow some of the several figures showing proportions to be replaced.
3) C. difficile is one of the microbes for which the epidemiology is difficult – the definition of HAI cases is not straightforward. IDSA recommends: “Use the same standardized case definitions (HO, CO-HCFA, CA) and rate expression (cases per 10 000 patient-days for HO). Ref. Practice Guidelines for Clostridium difficile Infection, CID 2018:66. It would be of interest to compare your HAI C. diff numbers with those of the community-acquired CDIs in your region. Please consider reporting this.
4) Acinetobacter baumanii as a very frequent pre-covid pathogen in pneumonia is somewhat surprising. The pathogen is an opportunist and, as C.diff, prevalent in hospitals with a high use of broad-spectrum antibiotics. You mention the antibiotic aspect in the article, but only in general terms, and no surveillance data on antibiotic use from Mures hospitals is being presented.
I suggest that you supply some hospital antibiotic use statistics, to better substantiate the cause of the high proportion of these pathogens.
5) if you apply prevalence rates instead of relative percentages, maybe the presence of A. baumanii, and the other non-covid pathogens for that matter, would appear unchanged (and even increasing) in the years 2020-21? If the decline is real, one may question if this may have been due to a marked shift in your patient cohorts during the pandemic. You address this shift in some detail in the lines 173-182. Please comment
6) I miss more specifics on the application of rigorous/uniform surveillance practices, and whether the registered data were approved by any infection control personnel after a common protocol. You cite the Romanian “Order No. 1101/2016” (ref. 5) which I have read. It does not seem to give such information, other than to cite the “Decision 2012/506/EU for the definition of HAIs” - indicating an alignment with European Union standards and guidelines. Other than that, it primarily focuses on the regulatory framework for managing HAIs and describes tasks and responsibilities. Interestingly, the “manager of the health unit” is “responsible for displaying on the unit's own website statistical information (quarterly and annual incidence rate, prevalence rate, quarterly and annual incidence broken down by types of infections and by wards) on healthcare-associated infections” – indicating that the abovementioned indices should be readily available. Regarding Clostridioides difficile, the Order emphasizes standard precautions and mandates the reporting and surveillance of HAIs. I find no specific mention of COVID-19 protocols, but such material might be included in other documents or annexes.
Could you comment on this important aspect, and if possible give some references to the HAI definitions used?
Discussion
1) The discussion is quite long, and I commend your different explanations for the rise in numbers of Covid and C.difficile cases; overcrowding, isolation difficulties etc.
However, you provide large text paragraphs to explain a possible pathogenetic mechanism (lines 200-210) to explain the overlapping (?) rise in C.diff and Covid. My question mark is inserted because these trends are only depicted in your Figure 6 while more robust data is lacking (see also comments on prevalence rates). Is it not probable that the common cause for this is the overuse of antibiotics that has been documented by several authors during the pandemic, in addition to overcrowding and insufficient infection control adherence?
2) In the lines 222-253, you describe various aspects of C. difficile management and treatment regimens that seem somewhat irrelevant to this paper's theme. I would suggest you minimize this or leave it out completely.
3) As you state in lines 293-314, neonates are a minor group (here 1.39% of HAIs). The lengthy text and the references pertaining to medical treatment and microbiology thus seem like a sidetrack in this study, which at the outset is epidemiological. I would rather suggest you reference and comment more upon the limitations of the study that prevail if you in the end do leave out prevalence rates, as commented upon earlier. It would be appropriate to compare your findings with those of authors, which is for a large part lacking.
For instance, studies have shown varying prevalence rates of Clostridioides difficile infections in different healthcare settings, influenced by antibiotic use and infection control practices (Cohen et al., 2010). Similarly, the prevalence of Acinetobacter in hospital settings has been linked to specific risk factors, as detailed in studies like Villar et al. (2013)
4) Except for the Covid-19 pandemic, did you have any other large ID outbreaks in your geographical region during the 5-year period which would maybe have skewed the data?
Figures and tables
Most Figures show the same relative percentages of HAIs versus various clinical and administrative parameters.
Figure 1: The percentage of the bars adds up to only 95%. I have not checked other figures.
Figure 5: I suggest the text is revised. What is the isolation and (in particular) the contact history, and is Yes = the patient was isolated / the patient had contact with a sick person in the hospital? How is a contact defined?
Comments on the Quality of English LanguageThe language of the manuscript is generally clear and formal, appropriate for an academic publication. However, there are minor issues:
Style: The writing style is mostly consistent with scientific literature. But some sentences are lengthy and complex, which might challenge readability.
Clarity: The manuscript is largely clear in its presentation of data and findings. However, simplifying complex sentences could definitively enhance understanding for a broader audience.
Author Response
REVIEW 2
The study focuses on hospital-acquired infections (HAIs) in public hospitals in Mures County, Romania. It is a cross-sectional study using data from 2017 to 2021, covering 4603 HAI cases. Key findings include:
- The most prevalent HAIs were enterocolitis with Clostridioides difficile, COVID-19, and bronchopneumonia.
- The study emphasizes the increasing trend of Clostridioides difficile cases, especially during the COVID-19 pandemic, and highlights the challenges posed by multi-resistant bacterial strains.
- It discusses the importance of appropriate antibiotic use and the need for continued vigilance in managing HAIs, particularly in the context of the pandemic.
- Of the 30% of patients evaluated, only 7% of deaths could be attributed to the HAI
The study's strengths lie in its comprehensive data collection over five years and its focus on a range of HAIs. However, the authors acknowledge limitations such as regional focus and incomplete data on patient comorbidities and risk factors.
I have several comments, which mainly relate to the cross-sectional method and the manuscript's scarcity of details on HAI definitions and data collection. Clear definitions and transparent reporting of data collection methodologies are crucial for reproducibility and reliability. I would encourage the authors to elaborate on the methods of data collection and analysis to bolster the study's credibility.
We express our gratitude for your active engagement and valuable suggestions for this manuscript. We highly appreciate initiatives dedicated to enhancing the quality of articles in specialized medical literature. Your suggestions, accompanied by clear and concise explanations, have served as valuable guidance, leading us to enhance the overall quality of this article. The implemented changes are highlighted in the manuscript in alignment with your recommendations:
Material and methods
The authors report a high proportion of HAIs caused by Clostridioides difficile, Covid-19 and Acinetobacter infections which is interesting, but to report the relative percentage of all HAIs as the only metric gives in my opinion relatively little information.
1) I suggest you include prevalence rates, which also aligns better with your use of cross-sectional and prevalence as your method, i.e., number of infections per year and population size. This would greatly improve the quality of this retrospective study. Standardizing the number of HAIs per total number of occupied hospital beds or by 1000 patient days, e.g. quarterly or per annum, is standard epidemiological practice. Such metrics would allow for more nuanced analyses, to benchmark between different hospitals, and better be able to study temporal trends.
Your suggestion was very well received. Prevalence rates have been incorporated into the manuscript to articulate the findings pertaining to HAIs during the examined time frame.
2) Regarding Covid-19 epidemiology, one may retrieve much information of interest from open sources. According to downloadable European Covid-19 data from the ECDC database, Romania in 2020-21 registered 1806554 new cases. The region of MureÊ‚ had a 2023 population of 518193 (Wikipedia), out of 19 million in Romania. This indicates a national prevalence per 1000 inhabitants of Covid-19 cases of 0,96 for the pandemic period (1806554 cases / 19000000 inhab * 1000) compared to the prevalence of nosocomial Covid-19 HAI cases which is on average 0,17 in Mures hospitals (913 HAI cases / 518193 inhab * 1000). This only 5-6-fold lower prevalence in hospitals than in the community seems high – but only a calculation of the prevalence rate using the number of bed-days or discharges from the hospitals as denominators, will allow comparison with international prevalence data.
I propose that this would improve the article and allow some of the several figures showing proportions to be replaced.
Your recommendation was taken into consideration and we made modifications regarding the prevalence rates as you suggested. In the manuscript, you will find in highlight that the prevalence of COVID-19 HAIs for the 2020–2021 time frame was calculated at 0.48 (913 cases/191880 discharges). Also, the prevalence of HAIs caused by SARS-COV2 calculated for the entire 5-year study was 0.17 (913 cases/530181 discharges). We added to the manuscript Table 2, in which we noted the prevalence of HAIs caused by each agent before and after the pandemic time, as well as the overall calculated prevalence for the study period.
3) C. difficile is one of the microbes for which the epidemiology is difficult – the definition of HAI cases is not straightforward. IDSA recommends: “Use the same standardized case definitions (HO, CO-HCFA, CA) and rate expression (cases per 10 000 patient-days for HO). Ref. Practice Guidelines for Clostridium difficile Infection, CID 2018:66. It would be of interest to compare your HAI C. diff numbers with those of the community-acquired CDIs in your region. Please consider reporting this.
Your suggestion regarding the comparison between HAI Cl. Difficile and community-acquired Cl. Difficile it would be in fact a very interesting subject to study. At the moment, we do not have data for the number of community-acquired Cl. Difficile, as this data only focused was HAIs from public hospitals in our region. For the future, we have in mind to study the comparison that you kindly suggested, using proper data regarding community-acquired infections.
4) Acinetobacter baumanii as a very frequent pre-covid pathogen in pneumonia is somewhat surprising. The pathogen is an opportunist and, as C.diff, prevalent in hospitals with a high use of broad-spectrum antibiotics. You mention the antibiotic aspect in the article, but only in general terms, and no surveillance data on antibiotic use from Mures hospitals is being presented.
I suggest that you supply some hospital antibiotic use statistics, to better substantiate the cause of the high proportion of these pathogens.
Regarding your advice to supply the manuscript with statistics about antibiotic use in our hospitals during this period, we would like you to know that this manuscript was written as the first part of a bigger perspective project, that has in plan to study antibiotic usage and antibiotic resistance our hospitals, especially regarding HAIs. Considering this aspect, we approached in general terms the aspect of antibiotic therapies. However, after reading your review, we made some changes in those paragraphs, wanting to improve the quality of the text according to your recommendations.
5) if you apply prevalence rates instead of relative percentages, maybe the presence of A. baumanii, and the other non-covid pathogens for that matter, would appear unchanged (and even increasing) in the years 2020-21? If the decline is real, one may question if this may have been due to a marked shift in your patient cohorts during the pandemic. You address this shift in some detail in the lines 173-182. Please comment
Your suggestion was very well received and we made the modifications accordingly. We used prevalence rates instead of relative percentages and this change showed that the non-covid pathogens were indeed increased in the years 2020-2021. This review point was the one that changed the results and discussion part (highlighted in the manuscript) of our paper and we would like to thank you for your important input. More than that, the results expressed in prevalence are in agreement with other results in the literature. We are grateful for your help in transmitting the most relevant and correct data.
6) I miss more specifics on the application of rigorous/uniform surveillance practices, and whether the registered data were approved by any infection control personnel after a common protocol. You cite the Romanian “Order No. 1101/2016” (ref. 5) which I have read. It does not seem to give such information, other than to cite the “Decision 2012/506/EU for the definition of HAIs” - indicating an alignment with European Union standards and guidelines. Other than that, it primarily focuses on the regulatory framework for managing HAIs and describes tasks and responsibilities. Interestingly, the “manager of the health unit” is “responsible for displaying on the unit's own website statistical information (quarterly and annual incidence rate, prevalence rate, quarterly and annual incidence broken down by types of infections and by wards) on healthcare-associated infections” – indicating that the abovementioned indices should be readily available. Regarding Clostridioides difficile, the Order emphasizes standard precautions and mandates the reporting and surveillance of HAIs. I find no specific mention of COVID-19 protocols, but such material might be included in other documents or annexes.
Could you comment on this important aspect, and if possible, give some references to the HAI definitions used?
We thank you for your interesting comment. For processing the presenting data, we received the consent of the Mures Public Health Directorate. Regarding the rigorous surveillance practices that you mention, our hospitals have several individual infection surveillance and reporting methodologies. The HAI definition used is the one specified in the Order. No. 1101, briefly detailed in the "HAI identification and diagnosis in Romania" section of the text.
Disscutions
1) The discussion is quite long, and I commend your different explanations for the rise in numbers of Covid and C.difficile cases; overcrowding, isolation difficulties etc.
However, you provide large text paragraphs to explain a possible pathogenetic mechanism (lines 200-210) to explain the overlapping (?) rise in C.diff and Covid. My question mark is inserted because these trends are only depicted in your Figure 6 while more robust data is lacking (see also comments on prevalence rates). Is it not probable that the common cause for this is the overuse of antibiotics that has been documented by several authors during the pandemic, in addition to overcrowding and insufficient infection control adherence?
Thank you for your useful review. The discussion part was revised and we made some changes in accordance with your remarks. After modifying the content of the results using your thoughtful suggestions written above regarding the use of prevalence data instead of relative percentages, we also revised the discussion part.
2) In the lines 222-253, you describe various aspects of C. difficile management and treatment regimens that seem somewhat irrelevant to this paper's theme. I would suggest you minimize this or leave it out completely.
Thank you for your suggestion, the part that you mentioned was removed from the manuscript after the revision of the discussion section.
3) As you state in lines 293-314, neonates are a minor group (here 1.39% of HAIs). The lengthy text and the references pertaining to medical treatment and microbiology thus seem like a sidetrack in this study, which at the outset is epidemiological. I would rather suggest you reference and comment more upon the limitations of the study that prevail if you in the end do leave out prevalence rates, as commented upon earlier. It would be appropriate to compare your findings with those of authors, which is for a large part lacking.
For instance, studies have shown varying prevalence rates of Clostridioides difficile infections in different healthcare settings, influenced by antibiotic use and infection control practices (Cohen et al., 2010). Similarly, the prevalence of Acinetobacter in hospital settings has been linked to specific risk factors, as detailed in studies like Villar et al. (2013)
Regarding the lines mentioned in the review above, we revised the entire discussion part. The lines that you mentioned earlier were removed. In the discussion part, we included more comparisons with those of authors, as you kindly suggested.
4) Except for the Covid-19 pandemic, did you have any other large ID outbreaks in your geographical region during the 5-year period which would maybe have skewed the data?
During the 5-year period studied, we did not have any other large ID outbreak in our geographical region.
Figures and tables
Most Figures show the same relative percentages of HAIs versus various clinical and administrative parameters.
Figure 1: The percentage of the bars adds up to only 95%. I have not checked other figures.
Figure 5: I suggest the text is revised. What is the isolation and (in particular) the contact history, and is Yes = the patient was isolated / the patient had contact with a sick person in the hospital? How is a contact defined?
We made modifications in accordance with your remarks – figure 1 was left out completely and we inserted Table 2 instead. This table contains the prevalence of HAIs caused by different etiological agents pre and post-pandemic, as well as for the entire 5-year period.
Figure 5 – We revised the figure, after revising the entire data and changing our focus on the prevalence as you suggested, we decided to leave out Figure 5.
Comments on the Quality of English Language
The language of the manuscript is generally clear and formal, appropriate for an academic publication. However, there are minor issues:
Style: The writing style is mostly consistent with scientific literature. But some sentences are lengthy and complex, which might challenge readability.
Clarity: The manuscript is largely clear in its presentation of data and findings. However, simplifying complex sentences could definitively enhance understanding for a broader audience.
We made revisions to the manuscript, incorporating the suggestions provided in your review. Specifically, we have enhanced the discussion section and simplified intricate sentences as per your recommendations. Your input has significantly elevated the quality of this manuscript. Thank you for your attention and patience as you guided us through the review of this article.
Round 2
Reviewer 2 Report
Comments and Suggestions for Authors
My comments have been and satisfactory answered and the manuscript revised accordingly.